# Aetiological Characteristics of Infectious Diarrhea in Yantai City, Shandong Province, China in 2017

**DOI:** 10.3390/v14020216

**Published:** 2022-01-23

**Authors:** Zhenlu Sun, Jinjie Xu, Peihua Niu, Miao Jin, Qiao Gao, Ruiqing Zhang, Ji Wang, Yong Zhang, Xuejun Ma

**Affiliations:** 1NHC Key Laboratory of Medical Virology and Viral Diseases, National Institute for Viral Disease Control and Prevention, Chinese Center for Disease Control and Prevention, Beijing 102206, China; sunluyu1007@126.com (Z.S.); peihua.niu@163.com (P.N.); hebmuzhangruiqing@163.com (R.Z.); 2Yantai Center for Disease Control and Prevention, Yantai 264003, China; xujinjie198205@126.com (J.X.); gaoqiao198402@126.com (Q.G.); 3Department of Viral Diarrhea, National Institute for Viral Disease Control and Prevention, Chinese Center for Disease Control and Prevention, Beijing 102206, China; jinmiao37@126.com; 4Chinese Field Epidemiology Training Program, Chinese Center for Disease Control and Prevention, Beijing 100050, China; 5WHO WPRO Regional Polio Reference Laboratory, NHC Key Laboratory for Biosafety, NHC Key Laboratory for Medical Virology, National Institute for Viral Disease Control and Prevention, Chinese Center for Disease Control and Prevention, Beijing 102206, China; 6Center for Biosafety Mega-Science, Chinese Academy of Sciences, Wuhan 430071, China

**Keywords:** infectious diarrhea, pathogen spectrum, bacteria, virus

## Abstract

This study aimed to analyse the pathogenic spectrum and epidemiological characteristics of infectious diarrhea in Yantai City, Shandong Province, China and provide a reference for its prevention and control. A total of 713 stool specimens collected within 3 days of diarrhea onset from January to December 2017 at secondary or higher hospitals in Yantai City were tested for 10 causative pathogens, using real-time polymerase chain reaction (RT-PCR). The top two rotaviruses and norovirus were analysed for typing and geographical distribution. The total positive rate was 46.56% (332/713), and 268 of 713 specimens contained at least one pathogen; 64 had at least two pathogens, accounting for 19.28% of the positive specimens (64/332). The positivity rates of rotavirus (RV), norovirus (NoVs) GI, norovirus (NoVs) GII, enterovirus universal (EV), enteric adenoviruses (EAdV), sapovirus (SaV), astrovirus (Astv), *Salmonella (SE), Listeria monocytogenes (LiMo), and Vibrio parahaemolyticus (VP)* were 20.06% (143/713), 1.82% (13/713), 12.84% (89/713), 10.66% (76/713), 4.07% (29/713), 0.42% (3/713), 2.38% (17/713), 1.54% (11/713), 1.82% (13/713), and 1.54% (11/713), respectively. Infectious diarrhea showed a high prevalence in young children aged 1-5 years, accounting for 48.6% of the total number of cases. Bacterial diarrhea was predominant in summer, and viral diarrhea was distributed throughout the year, without a significant seasonal pattern. Rotavirus is dominated by G9P, accounting for 81.82%, while norovirus is dominated by the GII type and has diverse characteristics. The aetiology of infectious diarrhea in Yantai is mainly viral, with RV, NoVs, EV, EAdV, and Astv being the most frequent pathogens. Continuous surveillance of infectious diarrhea diseases can help us understand its epidemiological and pathogenic characteristics, thereby taking targeted preventive and control measures in different seasons.

## 1. Introduction

Diarrhea is a global public health problem, causing approximately 760,000 child deaths worldwide each year [1]. It is classified by aetiology as infectious or non-infectious diarrhea. Infectious diarrhea is caused by bacteria, fungi, viruses, or parasites, in the intestinal tract, with diarrhea as the main clinical manifestation [2] and one of the major infectious diseases in both developed and developing countries. In China, infectious diarrhea has always been ranked among the highest incidences of statutory infectious diseases. Therefore, long-term surveillance of the pathogenic spectrum composition and epidemiological characteristics of infectious diarrhea can provide a basis for establishing prevention and control strategies and the adoption of targeted preventive and control measures. This study aimed to investigate the pathogenic spectrum of infectious diarrhea and the molecular epidemiological pattern of key causative pathogens in Yantai City, Shandong Province, China in 2017, and to provide a basis for improving relevant preventive measures and reducing the disease burden in this region.

## 2. Materials and Methods

### 2.1. Sample Source and Sample Collection

Clinical and laboratory diagnoses of infectious diarrhea were made according to the diagnostic criteria for infectious diarrhea (WS271-2007) [3]. When suspected cases of infectious diarrhea are found in medical institutions at all levels in 13 districts and counties under the jurisdiction of Yantai City, Shandong Province, they are reported directly through an online platform following the Regulations on the Management of Infectious Disease Information Reporting. Those without direct online reporting capacity report to the local county-level CDC within 24 h and send out the filled infectious disease report card within 24 h. The county-level CDC should investigate the first five cases within 48 h after receiving the case report (all cases should be investigated if there are fewer than five cases), and the “Case Investigation Form of Viral Diarrhea Surveillance Cases” should be completed.

Faecal specimens were collected from 713 patients with clinically suspected infectious diarrhea in Yantai City throughout 2017. Medical institutions at all levels in the 13 districts and counties under the jurisdiction of Yantai City were arranged by the local CDC to collect faecal specimens and conduct case investigations for the first five cases. None of the patients were administered with antibiotics prior to sampling. The collected specimens were refrigerated at 4 °C, sent to the laboratory within 24 h, and frozen at −80 °C for centralised testing every week.

### 2.2. Nucleic Acid Extraction and Detection

Total nucleic acid was extracted from 713 clinical specimens using faeces with the size of a mung bean or 50–100 μL aqueous faeces, adding 500 μL isotonic sodium chloride solution to make a 10–20% suspension, and then centrifuged at 8000 r/min for 5 min; 200 μL of the supernatant was collected, and the nucleic acid was extracted from the treated specimens.

Nucleic acid extraction was performed using an automatic nucleic acid extractor and corresponding kits from Sansure Biotech Inc. Pathogen detection was performed by real-time PCR (RT-PCR) on a Roche Light Cycler 480 II PCR using kits from Shanghai BioGerm Medical Technology Co., Ltd. (Shanghai, China) for 10 infectious diarrheal pathogens, including rotavirus (RV), norovirus (NoVs)GI, norovirus (NoVs)GII, enterovirus universal (EV), enteric adenoviruses (EAdV), sapovirus (SaV), astrovirus (Astv), Salmonella (SE), Listeria monocytogenes (LiMo), and Vibrio parahaemolyticus (VP). 

### 2.3. G/P Typing of Rotavirus

As RV is ranked first among infectious diarrheal pathogens in Yantai, two rounds of nested RT-PCR were used to type RV-positive specimens by amplifying the VP7 (G type) and VP4 (P type) gene regions. The primers and amplification conditions were as previously described [4]. Electrophoresis and digital analysis were performed using the QIAGEN OneStep RT-PCR Kit (QIAGEN GmbH, GERMANY) and QIAxcel capillary electrophoresis apparatus of 2xTaq PCR Starl MiX (Genstar Biochem, Canada). The G and P genotypes were determined based on the specific nucleic acid band size. 

### 2.4. Genotyping of Norovirus GII Specimens

As norovirus GII accounted for second place in the pathogen spectrum, GII-positive specimens were further genotyped to identify its dominant types. Molecular typing was performed on 40 GII-positive specimens with real-time PCR CT values of <30. NoV capsid protein region-specific oligonucleotide primers were used: COG2F:5′-CARGARBCNATGTTYAGRTGGATGAG-3;G2-SKR:5′-CCRCCNGCATRHCCRTTRTACAT-3’ [5]. A TaKaRa One-Step RNA PCR Kit (Dalian, China) was used. The reaction conditions were as follows: 30 cycles of incubation at 50 °C for 30 min, denaturation at 94 °C for 2 min, denaturation at 94 °C for 30 s, annealing at 42 °C for 30 s, and extension at 60 °C for 45 s, followed by a final cycle at 72 °C for 10 min. The electrophoresis products were confirmed as amplified fragments according to the DNA standard molecular mass (marker) position. Electrophoresis-positive samples were sent to Sangon Biotech (Shanghai, China) for nucleic acid sequence determination. The sequencing results were analysed using BioEdit software version 7.0 [6] for quality analysis and spliced using SeqMan combined in DNAStar software version 5.0 [7]. BLAST was used to retrieve the reference sequences after splicing. A phylogenetic tree based on the VP1 gene was drawn in MEGA 7.0 [7], using the neighbour-joining method with a bootstrap value set to 1000 replicates. 

### 2.5. Statistical Methods

Data were collated and analysed using Excel (version 2007) and SPSS (version 18.0, IBM Corp., Armonk, NY, USA), and statistical analysis was performed using the chi-square test, with *p* < 0.05 as a statistically significant difference. 

## 3. Results

### 3.1. Basic Information 

From 1 January to 31 December 2017, a total of 713 cases of diarrhea were registered in the comprehensive disease surveillance system in Yantai City, Shandong Province, China. The age of the patients was 0–89 years, 387 men and 326 women, the men to women ratio was 1:1.19.

### 3.2. Pathogenic Spectrum of Infectious Diarrhea

As shown in Figure 1, among the 713 faecal specimens, the total positivity rate was 46.56% (332/713). Two hundred sixty-eight specimens contained at least one pathogen, and 64 were infected with mixed pathogens, accounting for 19.28% (64/332) of the positive specimens. The positive detection rates for RV, Norovirus GI, Norovirus GII, EV, EAdV, SaV, Astv, SE, LiMo, and VP were 20.06% (143/713), 1.82% (13/713), 12.84% (89/713), 10.66% (76/713), 4.07% (29/713), 0.42% (3/713), 2.38% (17/713), 1.54% (11/713), 1.82% (13/713), and 1.54% (11/713), respectively. The positive specimens for SaV and Astv were all associated with mixed infections with other viruses.

### 3.3. Seasonal Distribution of Pathogens in Diarrhea Specimens

The pathogen spectrum showed clear seasonal variation (Figure 2). Viral infections are distributed throughout the year, with RV and Astv infection peaks occurring mainly in winter and spring. In contrast, EAdV had a high incidence in the summer, while EV and NoV infections have no specific time distribution characteristics. Bacterial infections were more seasonal than viral infections, showing a peak in summer, such as Limo and VP occurring in July-September.

### 3.4. Age Distribution of Patients with Infectious Diarrhea

The age of patients with diarrhea in this study was 0–89 years. They were divided into six groups: neonatal (0–1 year old), infant (2–5 years old), juvenile (6–18 years old), youth (19–30 years old), middle-aged (31–60 years old), and older groups (61–89 years old), as can be seen in Figure 3. The age of onset of infectious diarrhea was mainly concentrated in infants < 1-year-old, and 151 positive samples were detected in those < 1 year old, accounting for 45.48% (151/332) of the total population with diarrhea. The positive rates of RV, NoV, EV, LiMo, SE, and EAdV were significantly higher in patients aged less than 1 year than in other age groups. The incidence of infectious diarrhea was the lowest in the age group of 6–18 years, the pathogen combination was the most complex in the age group of 19–30 years, and the VP infection was concentrated in the 19–60 years age group.

### 3.5. Distribution of RV and NoVs in Each District of Yantai

Yantai city consists of 13 districts, and in this study, RV and NoV, the top two causative pathogens of infectious diarrhea, were analysed in each district. The results showed that the total RV and NoV infection rates in Zhifu, Penglai, and Haiyang districts, were significantly higher than those in other regions. (Figure 4).

### 3.6. Genotyping of RV-Positive Samples

A total of 143 RV-positive samples were genotyped by GP to understand their typing characteristics further. Among these, G9 was the predominant G genotype, accounting for 60.84% (87/143), followed by G2, G1, and G3, with 19.58% (27/143), 9.09% (13/143), and 4.19% (6/143), respectively. In addition, 3.50% (5/143) of the mixed G type and 2.10% (3/143) of the untyped G were found. The predominant P genotype was P [8], accounting for 75.52% (108/143), followed by P [4], 19.58% (28/143), mixed P, 2.78% (4/143), and untyped P, 5.59% (8/143). The G/P combination was predominantly G9P [8], accounting for 81.82% (117/143), followed by G2P [4], accounting for 10.49% (15/143).

### 3.7. Genotyping of GII Positive Samples

In total, 87 GII-positive specimens were obtained from Yantai in 2017. Molecular typing was performed by sequencing 40 GII positive specimens with real-time PCR CT values < 30. Due to partial nucleic acid degradation during sample sequencing or insufficient remaining samples, 19 GII positive sequences (387 bp in length) were obtained and compared with the published NoV sequences in GenBank. As shown in Figure 5, 19 sequences fell into the categories GII17, GII12, GII21, GII2, and GII4, indicating the diversity of GII types in Yantai. 

## 4. Discussion 

Etiological surveillance has important public health significance for the prevention and control of infectious diarrhea by identifying different pathogenic infections with similar clinical manifestations and tracing the composition and changes in the pathogenic spectrum. In this study, we reported the pathogenic spectrum and epidemiological characteristics of infectious diarrhea based on its etiological surveillance system in Yantai City in China in 2017.

In this study, 713 faecal samples from patients with diarrhea were selected and tested for 10 bacterial and viral pathogens. The results showed that the total positive rate was 46.56% (332/713), including 309 viral positives, 35 bacteria-positives and 64 specimens infected with mixed pathogens. Viral diarrhea accounted for 93.1% of positive samples of infectious diarrhea in Yantai city, which is similar to the results in other regions of China, with viral diarrhea accounting for 60–75% of infectious diarrhea [8,9]. The top three pathogens found in Yantai city were RV, NoVs, and EV, consistent with the monitoring results in other areas of China [10,11]. Our survey also showed that mixed infections with pathogens accounted for 19.28% of positive specimens. Mixed infections with different pathogens may have a synergistic effect, thus causing deterioration of diarrhea [12,13].

Regarding the seasonal patterns of various pathogens in this study, bacterial infections showed obvious seasonality, and the occurrence of bacterial pathogens, such as SE, LiMo, and VP peaked in summer and autumn, consistent with relevant research results [14,15]. Viral infections, on the other hand, were distributed throughout the year; RV and Astv infection peaked in winter and spring, and EAdV infection was more frequent in summer. Pathogen infection with a high incidence of diarrhea should be considered in different seasons [8,16,17].

The monitored patients of this study were all age groups, ranging from 0 to 89 years, and were divided into six groups according to age, which helps to understand the pathogenic composition of infectious diarrhea in all age groups in Yantai City, Shandong Province. In this study, we found that the 10 pathogen-positive patients were mainly concentrated in the 0 to 1 year old category, which was consistent with the domestic and foreign research results [18,19,20]. This situation may be related to the development of the infant’s immune system and feeding methods. Our study also found that there were age differences in VP prevalence, and its positive rate was highest in those 19–60 years old. This phenomenon might be related to the fact that VP contaminates shellfish, fish, shrimp, and other seafoods in the coastal area of Yantai, and young and middle-aged people tend to prefer consuming raw or semi-raw seafoods [21], while people of lower age and the elderly eat less. Moreover, older children and adults in China usually do not actively seek medical treatment when they have diarrhea and prefer to self-medicate. As China’s notifiable infectious disease reporting information system is a passive monitoring network, only those who seek medical treatment are included in the monitoring system. We assume that this is also one of the reasons for the high proportion of reported cases among younger children.

Among viral pathogens, RV and NoV are still the main causes of infectious diarrhea in Yantai, and these two pathogens accounted for 73.7% of the total positive specimens, which is consistent with surveillance results in other areas in China [18,20]. RV is an important pathogen that causes severe diarrhea in infants and young children. Because RV has strong variability and pathogenicity, and there is no specific drug, the development and promotion of a rotavirus vaccine is currently the primary measure for infection control [18,22]. Symptoms caused by NoVs are generally self-limiting, but outbreaks are of concern [23]. 

Both RV and NoV are prone to gene recombination, resulting in new variants or genotypes. Therefore, the investigation of genotypic distribution is of great significance for epidemiological studies of viruses and the development and application of vaccines [24,25]. The prevailing RV serotypes and genotypes worldwide are G1P8, G2P4, G3P8, and G4P8 [24]. Our study showed that the G9P8 rotavirus was the dominant strain in Yantai city, accounting for 92.31% of rotavirus-positive samples together with G2P4. Therefore, attention should be focused on these two dominant serotypes of RV. The NoVs types were GII17, GII12, GII21, GII2, and GII4, indicating that the GII types of diarrhea circulating are diverse, and continuous surveillance of GII types of diarrhea is necessary. In addition, our study reported that the regional distribution of RV and NoV had the highest incidence. The results showed significant differences between the different regions of Yantai. The total infection rates in Zhifu, Penglai, and Haiyang were significantly higher than those in the other regions. Zhifu district, the core area of Yantai city, is characterised by a dense population and a high degree of population flow. The Penglai district and Haiyang city are coastal areas, have convenient transportation, and their residents eat more raw seafood. Therefore, the high incidence of RV and NoV in these three regions may be related to the high number of cases, dense population, convenient transportation, high degree of population mobility, and dietary habits.

This study has some limitations. The research object was mainly diarrhea in clinics with a few severe cases and a limited number of 10 pathogens causing infectious diarrhea studied. Parasitic diarrhea was not included.

## 5. Conclusions

The prevention and control of bacterial diarrhea should be focused on in summer, and viral diarrhea should be paid attention to year-round, especially RV and NoV infections in Yantai city. Relevant departments should pay more attention to food hygiene and eating habits [26,27,28], reducing the incidence of diarrhea and the economic burden.

## Figures and Tables

**Figure 1 viruses-14-00216-f001:**
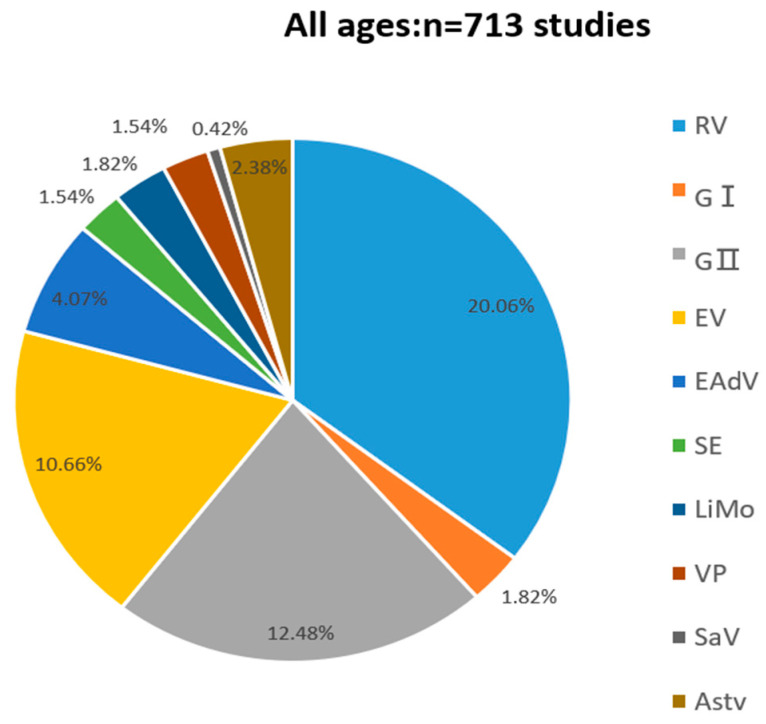
Pathogenic spectrum of infectious diarrhea in Yantai, China in 2017.

**Figure 2 viruses-14-00216-f002:**
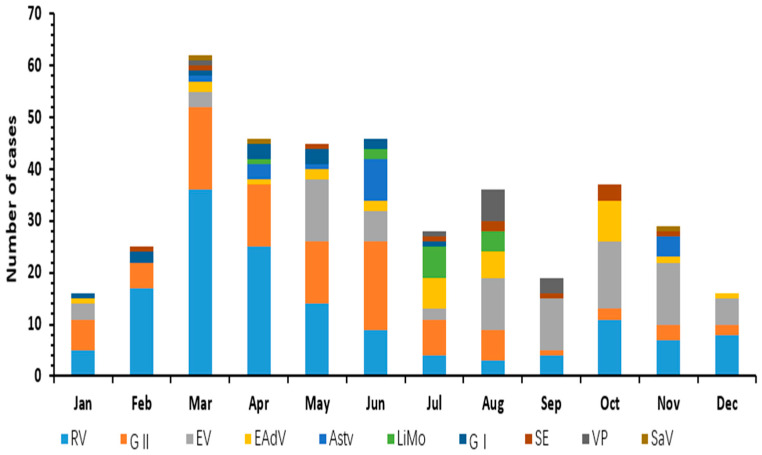
Seasonal distribution of ten pathogens of infectious diarrhea in Yantai, China, 2017.

**Figure 3 viruses-14-00216-f003:**
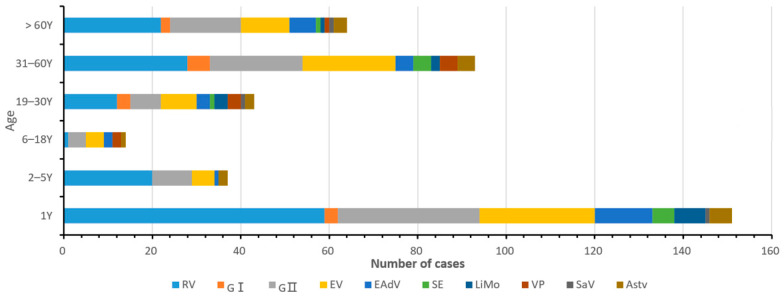
Age distribution map of ten pathogens of infectious diarrhea in Yantai, China in 2017.

**Figure 4 viruses-14-00216-f004:**
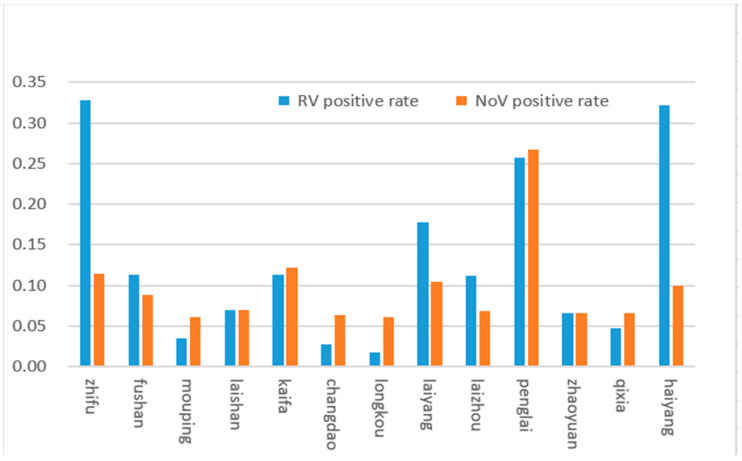
RV and NoVs distribution in different districts of Yantai in 2017.

**Figure 5 viruses-14-00216-f005:**
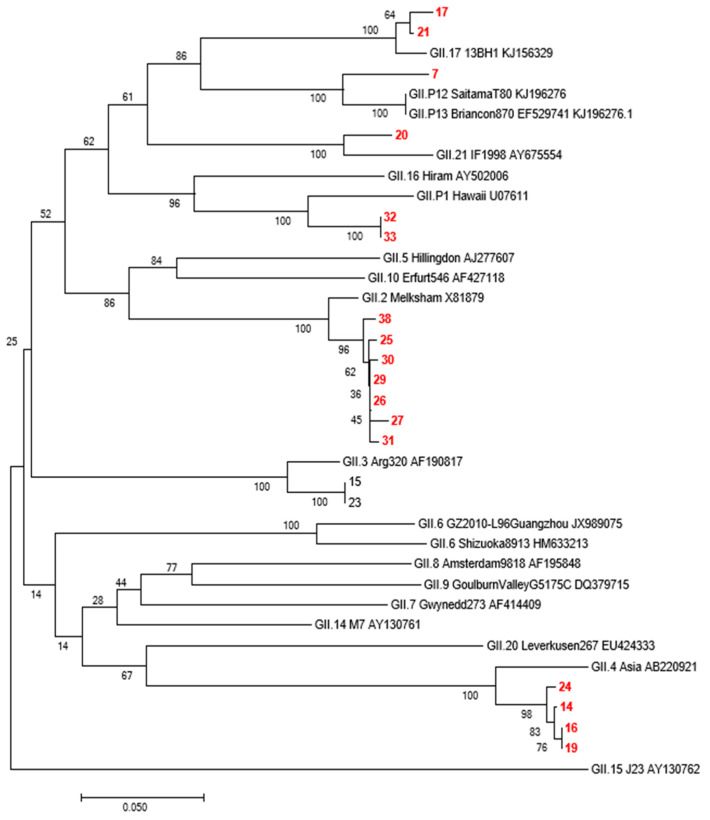
Genotyping and phylogenetic analysis of capsid protein region of Yantai Norovirus GII strain in 2017. Note: The red number represents the Yantai virus strain.

## Data Availability

Not applicable.

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
