# Peer review of "Aetiological Characteristics of Infectious Diarrhea in Yantai City, Shandong Province, China in 2017"

_viruses, 2022, doi:10.3390/v14020216_

Round 1

Reviewer 1 Report

General comments

The authors have examined infectious causes of diarrhea in Shandong province during the year 2017. Ten infectious pathogens were detected with a commercial assay. The detailed and interesting results are presented and visualized clearly.

A clinical interpretation of the data would enhance the study: E.g. Does the detected pathogen correlate to the disease course? Can that help establish an individual prognosis?

Line 166: “2.5. Distribution of RV and NoVs in Each District of Yantai” This data is not interpreted at all and thus of limited interest to a general audience. Please try to interpret the data, e.g. explain why RV infection rates are higher in the 3 named districts.

Line 183-185: “Molecular typing was performed by sequencing on 40 Gâ…¡positvie specimens with real-time PCR CT values below 30. Among them, 19 GII positive sequences (387 bp in length) were obtained” Why have only 19 sequences resulted from 40 GII positive specimens?

In line 210 and 211 you state: “mixed infection of different pathogens may have a synergistic effect, thus causing the de-210 terioration of diarrhea [12-13].” Can you confirm that in your own data?

Line 220: you state the oldest age was 89 years in line 223 you write that oldest age among diarrhea patients was 81 years old; please clarify!

Language and syntax

Line 154 to 157: “Each age group of 154 5 years was divided into 18 groups,as seen from Figure3, the age was mainly concentrated in children ≤5 years, with 187 out of 332 samples detected, accounting for 56.32%(187/332) 156 of the total population with diarrhea, which was statistically significant compared with 157 that of over 5 years group (χ2 = 242.94, P<0.05).” This sentence is too long and doesn´t flow well. I suggest you reformulate!

Line 191-193: “Figure 5. Phylogenetic analysis was carried out by genotyping and capsid protein region of Yantai norovirus GII strain detected from reference strains isolated worldwide.

Note: The virus strain shown is from Yantai red number”

Syntax and language errors! Please reformulate!

Line: 220: “each age group of 5 years was divided into 18 groups” This is misunderstanding, so please reformulate, e.g.  all patients were divided in age groups of 6  years, resulting in 18 groups

Formatting and typos

Line 108: “dominant types..Molecular” Delete one period

Line 184: “40 Gâ…¡positvie”

Line 271: “EAdV Enteric adeNoVsirus“; Please correct!

Line 267-279: Please format the abbreviations correctly, i.e. space between the columns

Please format bibliography uniformally!

Author Response

Response to Reviewer 1 Comments

Point 1: A clinical interpretation of the data would enhance the study: E.g. Does the detected pathogen correlate to the disease course? Can that help establish an individual prognosis?

Response to reviewer 1 comment 1: We appreciate your professional advice. However, the clinical data of disease course is not available in the present work, and the later work will strengthen the combination of pathogen detection results with clinical analysis.

Point 2: Line 166:“2.5. Distribution of RV and NoVs in Each District of Yantai”This data is not interpreted at all and thus of limited interest to a general audience. Please try to interpret the data, e.g. explain why RV infection rates are higher in the 3 named districts.

Response to reviewer 1 comment 2: We appreciate your professional advice. We have made relevant explanations and supplements in the manuscript.

The following: “The total infection rates in Zhifu, Penglai, and Haiyang were significantly higher than those in the other regions. Zhifu district, the core area of Yantai city, is characterised by a dense population and a high degree of population flow. The Penglai district and Haiyang city are coastal areas, have convenient transportation, and their residents eat more raw seafood. Therefore, the high incidence of RV and NoV in these three regions may be related to the high number of cases, dense population, convenient transportation, high degree of population mobility, and dietary habits.”(Page 8, lines 254-261)

Point 3:Line 183-185: “Molecular typing was performed by sequencing on 40 Gâ…¡positvie specimens with real-time PCR CT values below 30. Among them, 19 GII positive sequences (387 bp in length) were obtained” Why have only 19 sequences resulted from 40 GII positive specimens?

Response to reviewer 1 comment 3: Thanks for your advice. We have made relevant explanations and supplements in the manuscript.

The following: “Due to partial nucleic acid degradation during sample sequencing or insufficient remaining samples” (Page 6, lines 185-186)

Point 4:In line 210 and 211 you state: “mixed infection of different pathogens may have a synergistic effect, thus causing the deterioration of diarrhea [12-13].” Can you confirm that in your own data?

Response to reviewer 1 comment 4:Thanks for your advice. Since our current study was rarely combined with clinical data, there was no evidence in this regard. The results of other researchers provided us with this indication, and we will pay attention to the correlation between multi-pathogen infection and the severity of clinical symptoms in the later study.

Point 5:Line 220: you state the oldest age was 89 years in line 223 you write that oldest age among diarrhea patients was 81 years old; please clarify!

Response to reviewer 1 comment 5:Thanks for your advice. We have made corresponding changes in the manuscript. (Page 8, line 220)

Point 6:Line 154 to 157: “Each age group of 154 5 years was divided into 18 groups,as seen from Figure3, the age was mainly concentrated in children ≤5 years, with 187 out of 332 samples detected, accounting for 56.32%(187/332) 156 of the total population with diarrhea, which was statistically significant compared with 157 that of over 5 years group (χ2 = 242.94, P<0.05).” This sentence is too long and doesn´t flow well. I suggest you reformulate!

Response to reviewer 1 comment 6: Thanks for your advice.We have made relevant explanations and supplements in the manuscript.

The following:The age of patients with Diarrhea in this study was 0–89 years. They were divided into six groups: neonatal (0–1 years old), infant (2–5 years old), juvenile (6–18 years old), youth (19–30 years old), middle-aged (31–60 years old), and older groups (61–89 years old), as can be seen in Figure3. The age of onset of infectious Diarrhea was mainly concentrated in infants < 1-year-old, and 151 positive samples were detected in those < 1 year old, accounting for 45.48% (151/332) of the total population with Diarrhea. The positive rates of RV, NoV, EV, LiMo, SE, and EAdV were significantly higher in patients aged less than 1 year than in other age groups. The incidence of infectious Diarrhea was the lowest in the age group of 6–18 years, the pathogen combination was the most complex in the age group of 19–30 years, and the VP infection was concentrated in the 19–60 years age group. (Pages 4-5, lines 152-162)

Point 7:Line 191-193:“Figure 5.Phylogenetic analysis was carried out by genotyping and capsid protein region of Yantai norovirus GII strain detected from reference strains isolated worldwide.Note: The virus strain shown is from Yantai red number”,Syntax and language errors! Please reformulate!

Response to reviewer 1 comment 7: Thanks for your advice.We have made relevant modification in the manuscript. (Page 7, lines 193-195)

Point 8:Line: 220: “each age group of 5 years was divided into 18 groups” This is misunderstanding, so please reformulate, e.g. all patients were divided in age groups of 6 years, resulting in 18 groups.

Response to reviewer 1 comment 8: Thanks for your advice.According to the suggestion of another reviewer, we re-grouped the ages.

The following:The monitored patients of this study were all age groups, ranging from 0 to 89 years, and were divided into six groups according to age, which helps in understand the pathogenic composition of infectious Diarrhea in all age groups in Yantai City, Shandong Province. (Page 8, lines 220-222)

Point 9:Line 108: “dominant types..Molecular” Delete one period

Response to reviewer 1 comment 9: Thanks for your advice. We have modified it as suggested in the manuscript. (Page 3, line 107)

Point 10:Line 184: “40 Gâ…¡positvie”

Response to reviewer 1 comment 10: Thanks for your advice. We have modified it to“positive” in the manuscript. (Page 6, line 184)

Point 11:Line 271: “EAdV Enteric adeNoVsirus“; Please correct!

Response to reviewer 1 comment 11: Thanks for your advice. We have modified it to“Enteroadenovirus” in the manuscript. (Page 9, line 277)

Point 12:Line 267-279: Please format the abbreviations correctly, i.e. space between the columns.

Response to reviewer 1 comment 12: Thanks for your advice. We have modified it as suggested in the manuscript. (Page 9, lines 274-285)

Point 13:Please format bibliography uniformally!

Response to reviewer 1 comment 13:Thanks for your advice. We have modified it as suggested.

Response to Reviewer 1 Comments

Point 1: A clinical interpretation of the data would enhance the study: E.g. Does the detected pathogen correlate to the disease course? Can that help establish an individual prognosis?

Response to reviewer 1 comment 1: We appreciate your professional advice. However, the clinical data of disease course is not available in the present work, and the later work will strengthen the combination of pathogen detection results with clinical analysis.

Point 2: Line 166:“2.5. Distribution of RV and NoVs in Each District of Yantai”This data is not interpreted at all and thus of limited interest to a general audience. Please try to interpret the data, e.g. explain why RV infection rates are higher in the 3 named districts.

Response to reviewer 1 comment 2: We appreciate your professional advice. We have made relevant explanations and supplements in the manuscript.

The following: “The total infection rates in Zhifu, Penglai, and Haiyang were significantly higher than those in the other regions. Zhifu district, the core area of Yantai city, is characterised by a dense population and a high degree of population flow. The Penglai district and Haiyang city are coastal areas, have convenient transportation, and their residents eat more raw seafood. Therefore, the high incidence of RV and NoV in these three regions may be related to the high number of cases, dense population, convenient transportation, high degree of population mobility, and dietary habits.”(Page 8, lines 254-261)

Point 3:Line 183-185: “Molecular typing was performed by sequencing on 40 Gâ…¡positvie specimens with real-time PCR CT values below 30. Among them, 19 GII positive sequences (387 bp in length) were obtained” Why have only 19 sequences resulted from 40 GII positive specimens?

Response to reviewer 1 comment 3: Thanks for your advice. We have made relevant explanations and supplements in the manuscript.

The following: “Due to partial nucleic acid degradation during sample sequencing or insufficient remaining samples” (Page 6, lines 185-186)

Point 4:In line 210 and 211 you state: “mixed infection of different pathogens may have a synergistic effect, thus causing the deterioration of diarrhea [12-13].” Can you confirm that in your own data?

Response to reviewer 1 comment 4:Thanks for your advice. Since our current study was rarely combined with clinical data, there was no evidence in this regard. The results of other researchers provided us with this indication, and we will pay attention to the correlation between multi-pathogen infection and the severity of clinical symptoms in the later study.

Point 5:Line 220: you state the oldest age was 89 years in line 223 you write that oldest age among diarrhea patients was 81 years old; please clarify!

Response to reviewer 1 comment 5:Thanks for your advice. We have made corresponding changes in the manuscript. (Page 8, line 220)

Point 6:Line 154 to 157: “Each age group of 154 5 years was divided into 18 groups,as seen from Figure3, the age was mainly concentrated in children ≤5 years, with 187 out of 332 samples detected, accounting for 56.32%(187/332) 156 of the total population with diarrhea, which was statistically significant compared with 157 that of over 5 years group (χ2 = 242.94, P<0.05).” This sentence is too long and doesn´t flow well. I suggest you reformulate!

Response to reviewer 1 comment 6: Thanks for your advice.We have made relevant explanations and supplements in the manuscript.

The following:The age of patients with Diarrhea in this study was 0–89 years. They were divided into six groups: neonatal (0–1 years old), infant (2–5 years old), juvenile (6–18 years old), youth (19–30 years old), middle-aged (31–60 years old), and older groups (61–89 years old), as can be seen in Figure3. The age of onset of infectious Diarrhea was mainly concentrated in infants < 1-year-old, and 151 positive samples were detected in those < 1 year old, accounting for 45.48% (151/332) of the total population with Diarrhea. The positive rates of RV, NoV, EV, LiMo, SE, and EAdV were significantly higher in patients aged less than 1 year than in other age groups. The incidence of infectious Diarrhea was the lowest in the age group of 6–18 years, the pathogen combination was the most complex in the age group of 19–30 years, and the VP infection was concentrated in the 19–60 years age group. (Pages 4-5, lines 152-162)

Point 7:Line 191-193:“Figure 5.Phylogenetic analysis was carried out by genotyping and capsid protein region of Yantai norovirus GII strain detected from reference strains isolated worldwide.Note: The virus strain shown is from Yantai red number”,Syntax and language errors! Please reformulate!

Response to reviewer 1 comment 7: Thanks for your advice.We have made relevant modification in the manuscript. (Page 7, lines 193-195)

Point 8:Line: 220: “each age group of 5 years was divided into 18 groups” This is misunderstanding, so please reformulate, e.g. all patients were divided in age groups of 6 years, resulting in 18 groups.

Response to reviewer 1 comment 8: Thanks for your advice.According to the suggestion of another reviewer, we re-grouped the ages.

The following:The monitored patients of this study were all age groups, ranging from 0 to 89 years, and were divided into six groups according to age, which helps in understand the pathogenic composition of infectious Diarrhea in all age groups in Yantai City, Shandong Province. (Page 8, lines 220-222)

Point 9:Line 108: “dominant types..Molecular” Delete one period

Response to reviewer 1 comment 9: Thanks for your advice. We have modified it as suggested in the manuscript. (Page 3, line 107)

Point 10:Line 184: “40 Gâ…¡positvie”

Response to reviewer 1 comment 10: Thanks for your advice. We have modified it to“positive” in the manuscript. (Page 6, line 184)

Point 11:Line 271: “EAdV Enteric adeNoVsirus“; Please correct!

Response to reviewer 1 comment 11: Thanks for your advice. We have modified it to“Enteroadenovirus” in the manuscript. (Page 9, line 277)

Point 12:Line 267-279: Please format the abbreviations correctly, i.e. space between the columns.

Response to reviewer 1 comment 12: Thanks for your advice. We have modified it as suggested in the manuscript. (Page 9, lines 274-285)

Point 13:Please format bibliography uniformally!

Response to reviewer 1 comment 13:Thanks for your advice. We have modified it as suggested.

Reviewer 2 Report

Overall, Sun et al examine the causes of diarrheal diseases in a province within China throughout 2017.  Their methods are sound and did generate data that is informative from a regional perspective.  There is significance in the genotyping of the viruses, however, this epi data is almost 5 years old at this point and may have changed since the time of collection till now.  Overall, it is a descriptive work that may be published in this journal, though it might be more suited in a different broader based public health journal, as the authors do examine several bacterial species that cause infectious diarrhea.  The virology aspects do add regional significance for the genotyping.  The manuscript does need extensive editing for English grammar and punctuation.  For figure 3, much of the data looks spliced together and that figure could be cleaned up to make the data bars look more congruent.  

Author Response

Response to Reviewer 2 Comments

Point 1:The manuscript does need extensive editing for English grammar and punctuation.

Response to reviewer 2 comment 1:Thanks for your advice. We have modified it as suggested.

Point 2:For figure 3, much of the data looks spliced together and that figure could be cleaned up to make the data bars look more congruent.

Response to reviewer 2 comment 2:Thanks for your advice. We have modified it as suggested (Page 5 line 163).
